# Tau Isoforms: Gaining Insight into *MAPT* Alternative Splicing

**DOI:** 10.3390/ijms232315383

**Published:** 2022-12-06

**Authors:** Andrea Corsi, Cristina Bombieri, Maria Teresa Valenti, Maria Grazia Romanelli

**Affiliations:** Department of Neurosciences, Biomedicine and Movement Sciences, University of Verona, 37134 Verona, Italy

**Keywords:** Tau, RBP, alternative splicing, MAPT, PTBP1, tauopathies

## Abstract

Tau microtubule-associated proteins, encoded by the *MAPT* gene, are mainly expressed in neurons participating in axonal transport and synaptic plasticity. Six major isoforms differentially expressed during cell development and differentiation are translated by alternative splicing of *MAPT* transcripts. Alterations in the expression of human Tau isoforms and their aggregation have been linked to several neurodegenerative diseases called tauopathies, including Alzheimer’s disease, progressive supranuclear palsy, Pick’s disease, and frontotemporal dementia with parkinsonism linked to chromosome 17. Great efforts have been dedicated in recent years to shed light on the complex regulatory mechanism of Tau splicing, with a perspective to developing new RNA-based therapies. This review summarizes the most recent contributions to the knowledge of Tau isoform expression and experimental models, highlighting the role of *cis*-elements and ribonucleoproteins that regulate the alternative splicing of Tau exons.

## 1. Introduction

Tau is a microtubule-associated protein, transcribed by the microtubule-associated protein Tau (*MAPT*) gene, which plays a pivotal role in tubulin assembly and cytoskeleton stabilization [1]. *MAPT* mRNA is extensively regulated by alternative splicing (AS), leading to the expression of at least six main isoforms ranging from 352 up to 441 amino acids (aa). Tau proteins are differentially expressed in the nervous system depending on neuron type and maturation and participate in the regulation of morphogenesis, differentiation, and axonal transport [2]. Deregulation in Tau isoform expression and post-translational modifications contribute to neuron degeneration, leading to the formation of Tau aggregation and neurofibrillary tangles, which characterize tauopathies [3,4].

Tau isoforms are abundantly expressed in neuronal cells, but they are also distributed in many other tissues, although in smaller amounts. Several reports have described Tau expression deregulation in different types of cancer, such as breast, ovary, prostate, and gastric cancer [5,6,7,8,9]. Furthermore, Tau aggregation has been found in the muscles of patients diagnosed with inclusion body myositis (IBM) [10,11]. Additional roles other than mediating microtubule polymerization and stabilization have been attributed to Tau, including regulation of signal transduction and axonal transport, protecting genomic architecture, RNA metabolism, and protein synthesis [12,13].

The two major mechanisms that have been investigated in pathologies involving deregulated Tau expression are the unbalance in the expression of Tau isoforms and their aggregation and the main post-translational modification due to Tau phosphorylation, which have been extensively described in several exhaustive reviews [14,15,16,17,18]. Over the years, a great deal of effort has been dedicated to the investigation of the regulation of Tau mRNA splicing patterns, contributing to the identification of *cis*-acting sequences and splicing factors required for Tau transcripts expression and opening a new perspective in the development of molecular target therapeutic tools.

In this review, we will resume the most recent contributions to the knowledge of the expression and function of Tau isoforms by analyzing data from literature and bioinformatics databases. Special attention will be given to the splicing mechanism that regulates *MAPT* gene expression, with an emphasis on *cis*-determinants and *trans*-factors (i.e., the ribonucleoproteins) involved in the AS of the *MAPT* gene transcripts.

## 2. *MAPT* Gene Organization and Alternatively Spliced Transcripts

The human *MAPT* gene is located on chromosome 17q21 and is organized into 16 exons conventionally numbered as represented in Figure 1 [1,19,20,21]. Tau belongs to a family of homologous proteins, including MAP2 and MAP4, with 3 or 4 repeated basic microtubule-binding domains (MTBDs) in their carboxy-terminal regions. The splicing of at least six *MAPT* exons (2, 3, 4a, 6, 8, and 10) is regulated by AS, resulting in the production of more than a dozen different Tau protein isoforms [2,22]. Two genes internal to *MAPT* have been described: the *MAPT-IT1* (long non-coding RNA intron 1 transcript) and the saitohin (*STH*) single exon open-reading frame in *MAPT* intron 11 [23]. The *STH* protein has been demonstrated to interact with Tau and the non-receptor tyrosine kinase c-Abl (Abl) [24]. An accurate phylogenetic and comparative genomics analysis has highlighted the distinct genomic evolution of *MAPT* in vertebrates. Notably, *Homo sapiens MAPT* shares the greater homology with its orthologues in primates (the most similar being the *Hylobates lar,* the common gibbon) and in mammals in general [23]. *MAPT* orthologues in birds, reptiles, and amphibians share a certain degree of similarity, while fish display the lowest homology to the human *MAPT* protein among vertebrates [23]. Human and murine Tau genes (*MAPT* and *Mapt*, Figure 1) share the same intron-exon organization, albeit they differ slightly in their total size while the human *MAPT* gene spans 133.9 kb, the mouse orthologue appears to be slightly smaller, spanning 102.8 kb [25]. These orthologues also share similar GC content (46.9% for the human gene and 47.2% for the murine gene) and a strong similarity of the mRNA sequences. The exon sequences of the longest Tau isoform are 83% identical between the two species [25]. Human and murine Tau partially share the constitutive retention pattern of exons 1, 4, 5, 7, 9, 11, 12, and 13, all of which are always present in the mature mRNA (Figure 1, in indigo) [1,22]. The 5′ untranslated region (5′ UTR) extends from exon 0 to the first portion of exon 1 (218 bps in mice and 224 bps in humans) and displays a great degree of homology between humans and mice [1,25,26,27,28] (Figure 1, in gray). On the other hand, 3′UTR composition differs significantly between species. In mouse, rat, and cow mRNAs, the short intronic sequence between E13 and E14 can be eliminated during the splicing process [29,30], resulting in the truncation of the 3′-UTR and E14 incorporation in the mature mRNA (Figure 1, light blue) [31]. On the contrary, in humans, the intron between E13 and E14 is always retained, resulting in a long 3′-UTR spanning at least 2 kb [25,31,32].

Tau exons 2, 3, 4a, 6, 8, and 10 are alternatively spliced [21,22,26] using a general mechanism of inclusion/exclusion. Six main *MAPT* transcript variants are generated by the AS of exons 2, 3, and 10 and translated into different Tau protein isoforms expressed in the adult human central nervous system (CNS). Exon 2 inclusion is regulated during human development, being excluded during embryonic development and included during tissue differentiation in the CNS [26,33,34,35,36]. In contrast with exon 2, exon 3 expression is preferentially regulated via splicing exclusion by several regulatory elements [26,35,37]. Its preference for exclusion is primarily due to the presence of a weak branch point in the upstream intron, which favors exon 3 removal from the mature RNA [37]. In addition to this, E3 contains two splicing silencers and one single splicing enhancer, which participate in exon 3 regulation by cooperating with exon 2 sequences [26,37]. In fact, exon 3 inclusion occurs only if exon 2 is retained in the mature mRNA, acting as an activator of exon 3 inclusion [1,37].

Exon 4A is the biggest of the Tau exons, spanning slightly more than 1 kb (1065 bp as for Ensembl MAPT-201/ENST00000262410.10), originally identified in adult PNS (peripheral nervous system) and in retinal transcripts [33,34,38]. It displays a default splicing pattern of exclusion [29], and it may require an activator in order to achieve its inclusion [26], although the exact splicing mechanism is still to be determined.

Tau exon 6 expression was not observed in mammal transcripts up until the early 90s, when it was detected in both murine neuroblastoma and rat pheochromocytoma cell lines [33,34]. A year later, Andreadis and colleagues demonstrated that exon 6’s eventual inclusion in Tau mature mRNA is dependent on its flanking exon regions and that it cannot be considered a cassette [29]. Besides the use of the canonical 3′ site, exon 6 inclusion may also occur using two alternative 3′ splice sites, originally considered cryptic, named “p” (proximal) and “d” (distal), according to their position with respect to the beginning of exon 6 [29,39]. In humans, exon 6 inclusion (6+ isoforms) in the CNS is less represented, but it is extensively expressed both in spinal cord and skeletal muscle samples [40].

As regards exon 8, until now it has never been demonstrated to be transcribed in the human mature *MAPT* mRNA [22,31]. Exon-trapping assays demonstrate that exclusion is its default splicing pattern [29]. However, E8 inclusion in mature mRNA has been described in cows and rhesus monkeys [26,41,42,43] and mouse transcripts containing both E8 and E10 (Mapt-205/ENSMUST00000106993.10) are reported in the Ensembl database.

The complexity of *MAPT* gene organization is further increased by the presence, in intron 9, of the complete opening reading frame of the saitohin (*STH*) protein, which is highly conserved only in primates closely related to humans, such as chimpanzees, baboons, and gorillas [22,24,44].

The regulation of exon 10 expression is the most investigated splicing mechanism of *MAPT* transcripts due to its association with neurodegenerative diseases [15]. In healthy adult humans, inclusion and exclusion of E10 are balanced at a 1:1 rate in the CNS transcripts, whereas a shift in this equilibrium is associated with neurodegenerative disorders [26,45,46,47]. In humans, E10 expression requires developmental stage-specific splicing inhibitors, which lead to its exclusion in fetal mature RNAs and alternative regulation in adult tissues [26,48]. Interestingly, while E10 inclusion in the adult human brain is fine-tuned and balanced with its exclusion, in adult rodents, E10 is constitutively included in the mature RNA [49]. It has been demonstrated that in humans, the self-complementarity between the 3′ end of E10 and the 5′ end of I10 leads to the formation of a stem-loop at the exon-intron interface, while in rodents, the loop is destabilized because a nucleotide-base A at position E10+13 is replaced by a nucleotide-base G, hindering exon skipping [50].

Recently, Garcìa-Escudero and colleagues detected a novel *MAPT* mRNA species retaining part of intron 12 (TIR-MAPT, *Truncated by Intron Retention MAPT*) both in human neuroblastoma cells (SH-SY5Y line) and in human brain RNA samples [51]. Interestingly, the authors also detected a significant increase in the TIR-MAPT/total *MAPT* mRNA ratio when comparing Alzheimer’s Disease Braak Stage VI patients to healthy controls [51], highlighting that this novel mRNA species could be strongly correlated with neurodegenerative processes.

## 3. Tau Isoforms

Tau was first described as a natively unfolded microtubule-associated protein. In fact, its main function is to promote the assembly of microtubules and stabilize their structure [13]. However, Tau proteins have a variety of other functions, which include maintaining the structural integrity of neurons, contributing to signal transmission between neurons, and axonal transport [1,13]. Tau also plays a role in regulating myelination, iron homeostasis, and neurogenesis [13] and may also support synaptic plasticity [52]. Other roles attributed to Tau are gene expression regulation [53], DNA protection [54,55,56], genome stability [54], microRNA activity [57], RNA protection [56], RNA metabolism, and protein synthesis [12].

Tau knock-out mice developed deterioration of cardiovascular function, glucose intolerance, pancreatic disorders, anxiety, and impairment of contextual and cued fear memory, implying a wide range of undiscovered functions of Tau [58,59,60]. The regional distribution of mRNA expression and total Tau protein expression levels were largely in agreement, appearing to be highly correlated [61].

Tau is widely distributed in the nervous system. Its main localizations are peripheral nerves and the brain, but this protein has also been found in other organs or tissues, such as the salivary glands, pancreas, breast, kidney, testes, myocardium, and skeletal muscle (https://www.genecards.org, accessed on 2 October 2022), suggesting other functions for Tau that are not limited to the CNS [13]. The subcellular distribution of Tau is regulated during development, becoming enriched in axons as neurons establish their polarity via mechanisms that may include isoform specificity, local synthesis of Tau in axons, preferential binding of Tau to axonal microtubules, and/or preferential axonal transport of the protein [62].

In the brain, under physiological conditions, Tau is mainly present in the axons of neurons and at low levels in glia (both astrocytes and oligodendrocytes) [63,64], but it has also been detected outside the cells [1]. Tau is continuously secreted both enclosed within extracellular vesicles, such as exosomes or autophagosomes, and using non-vesicle-mediated pathways and direct secretion through the plasma membrane. This could indicate a functional role of extracellular Tau in neuronal activities that has not yet been physiologically clarified. Conversely, under pathological conditions, this mechanism contributes to the spread of altered forms of Tau across different brain areas and thus to the spatio-temporal disease progression in tauopathies [65,66].

### 3.1. Functional Domains Characterizing Tau Protein

Tau protein is organized into four main functional domains: the N-terminal region (NTR), the proline-rich region (PRR), the microtubule-binding domain (MTBD), and the C-terminal assembly region [1,22]. The organization of the functional domains of the main Tau isoforms is described in Figure 2.

Each different region present in the final protein has been demonstrated to be related to specific functions, while there is a considerable gap in the literature regarding the specific function of each Tau isoform. However, Tau-specific functions can be potentially derived based on the specific domains that are included in each isoform [22,67]. It is important to note that these domains are not functionally independent, but many of their functions overlap and are also dependent on the intramolecular interactions between them [68].

The *N-terminal region (NTR)* encompasses the beginning of the Tau protein until the first residue of exon 7 (aa 1–150; numbering refers to the 441aa CNS isoform). The structure of this domain is shown in Figure 2, where N1 represents the segment translated by exon 2 and N2 the segment translated by exon 3. NTR is also called “Projection domain”, as it projects away from the microtubule to interact with other cytoskeletal and cytoplasmic proteins or organelles, such as annexins, synaptic vesicle-associated proteins, or mitochondria [26,68,69]. NTR may regulate the subcellular distribution of Tau into axons and may also have a role in signaling and Tau aggregation [62,70,71]. Tau has been demonstrated to bind to DNA and RNA, as well as interact with chromatin and within the inner side of the nuclear lamina. Since the alternatively spliced isoforms including exon 2 are the only ones specifically targeted towards the nucleus, it was suggested that a portion of NTR encoded by exon 2 is responsible for these nuclear interactions of Tau [22]. Furthermore, N-terminally truncated Tau shows an increased association with microtubules. In vitro experiments revealed that two eight amino acid-long motifs within exon 1, which are conserved in mammals, are needed for the interaction with annexins A2 and A6. The lack of these amino acid segments moderately increased the Tau association rate to microtubules, consistent with the supposition that the presence of the Tau-annexin interaction reduces the availability of Tau to interact with microtubules [70]. Finally, when Tau is attached to microtubules, the projection domain regulates the spacing between microtubules in the axon and, therefore, the axonal diameter. Large diameter reduces the resistance of the axoplasm, helping axonal transport, particularly in peripheral neurons [62,69].

The *proline-rich region (PRR)* is codified by exon 7 and part of exon 9 (aa 151–243). The high proportion of proline residues (more than 20% higher than the average of human proteins) confers to this region an increased rigidity and basicity [22,68]. This region interacts with many kinases and phosphatases and seems to be involved in the regulation of Tau phosphorylation and in promoting microtubule assembly and stability with MTBD. The PRR is also involved in interactions with proteins containing a Src homology-3 (SH3) domain, playing an important role in protein folding [68,71,72,73]. Gene-ontology analysis revealed that the group of proteins that specifically interact with the PRR showed a remarkable enrichment of proteins involved in cell signaling mechanisms [68].

Regulation of affinity between Tau and microtubules mainly depends on phosphorylation of Tau, which mostly targets the proline-rich region and C-terminus [59,74]. PRR indeed has the highest relative content of serine and threonine, making it Tau’s prime region for phosphorylation. Both the interactions of the NTR and the MTBD can be affected by phosphorylation within the PRR, suggesting that the PRR can act as a signaling module for the functions of both the NTR and the MTBD [68].

The *microtubule-binding domain (MTBD)* ranges from the rest of exon 9 to exon 12 (aa 244–368). Each exon encodes one of the four highly conserved repeats of 18 aa, each separated by 13 or 14 residues (R1-R4 in Figure 2) [68,69]. The size of the MTBD in different Tau isoforms depends on the alternative splicing of exon 10, which is translated into the second repeat (R2 segment). The domain with the longest size, which contains all four repeats, shows a higher affinity for microtubule binding, in contrast to that lacking an R2 segment [1,28,69].

MTBD is responsible for Tau binding with microtubules through the interaction with microtubular tubulin, thus promoting microtubule assembly and stability [59]. Being positively charged, the MTBD can also easily make electrostatic interactions with the negatively charged C-terminal region, which seems to be essential for the extremely dynamic “kiss-and-hop” interaction (rapid binding and detaching from the microtubule surface, with a mean dwell time of single Tau molecules in the millisecond range) between microtubules and Tau. This mechanism was described for axonal microtubules in cultured neurons and seems to be essential for axonal transport [68,71,75]. Moreover, this region also interacts with other proteins, such as actin or heat shock proteins [76,77]. Due to its interaction with microtubules, Tau is a modulator in a number of cellular processes involving cytoskeletal structure, such as morphogenesis, cellular division, and intracellular trafficking of organelles and vesicles [68,69]. MTBD is also responsible for Tau self-aggregation and polymerization into filaments and neurofibrillary tangles, the typical lesions found in tauopathies [69,78,79].

Several proteins associated with neurodegenerative disorders interact with the MTBD and PRR domains of Tau, suggesting that the structural composition and protein–protein interactions of Tau play an important role in pathological processes [80].

The *C-terminal region*, translated by exon 13 (aa 369–441) contributes to microtubule binding and interacts with the N-terminus in the paperclip conformation crucial for Tau functions. The Tau molecule shows a preference for changing its global conformations to form a “paperclip” shape, in which the N-terminal, C-terminal, and repeat domains all approach each other [81]. The paperclip conformation is proposed to be a usual conformation of soluble Tau and crucial for Tau physiological functions, to protect Tau from aggregation. Modifications, such as phosphorylation or truncation on either end, can greatly disturb or prevent the formation of this structure and might thereby promote Tau aggregation [1,22].

Alternative splicing events lead to the extension, reduction, or deletion of Tau domains, as described in the following sections.

### 3.2. Central Nervous System (CNS) Isoforms

In the adult CNS, six major Tau isoforms of 352–441 aa and 37–46 kDa molecular mass are expressed, deriving from the alternative splicing of exons 2, 3, and 10 (Figure 2) [26,28]. According to the inclusion or exclusion of exon 10, which encodes for the R2 segment, the isoforms are labeled “4R-” or “3R-”, respectively. The inclusion or exclusion of exons 2 and 3, which encode for the N1 and N2 segments, is indicated in the second half of the isoform name as: “0N” (absence of both exons), “1N” (inclusion of exon 2 only), or “2N” (inclusion of both exons 2 and 3) [1]. 

In the adult human brain, the expression pattern of the *MAPT* gene is consistent with the production of an equimolar ratio of 4R and 3R CNS isoforms, while 0N, 1N, and 2N isoforms are differently expressed as 37%, 54%, and 9% of total Tau, respectively [1]. The balance between 4R and 3R isoforms is relevant for human brain function, as the splicing dysregulations of exon 10 were associated with tauopathies [28,82]. Differently from humans, adult mouse brain almost exclusively expresses the 4R isoforms, while the 3R isoforms are only transiently expressed in mouse newborn neurons [1]. Human *MAPT* gene expression is under temporal regulation. The above described landscape with all six CNS isoforms is found in the adult brain, whereas the 3R0N hyperphosphorylated isoform is predominant in the fetal brain, with a marked shift in exons 2 and/or 10 expression in the perinatal period [22,28,82,83].

### 3.3. Big-Tau

A high-molecular-weight isoform of about 110 kDa named “Big-Tau” was discovered in the 1990s. This isoform is predominantly expressed in adult PNS and can also be found in the optic nerve, in neurons of the CNS extending their processes into the periphery, and in PC12, a rat cell line derived from neural crest [62]. Big-Tau arises from a *MAPT* transcript of about 8–9 kb. This mRNA includes a large additional exon (encoding 254 aa in rats and 237 aa in mice) between exons 4 and 5, which was called 4a [33,34], resulting in a protein doubled in size with a great extension of the N-terminal projection domain (from 198 to 510 aa). Moreover, in some of the neural rodent cell lines that have been analyzed, the sequence of Big-Tau also includes exon 6 [62].

Most of the available information about Big-Tau derives from studies on mouse and rat models as well as from cell lines derived from them. Data on Big-Tau isoforms from humans and other species have been derived mostly from genomic analysis based on transcript alignments [62]. Big-Tau expression in rodents begins late in embryonic development and gradually increases postnatally [34,84]. The functional reason for the switch from a low-molecular-weight isoform to Big-Tau in specific neuronal populations is still unknown. Oblinger’s group speculated that Big-Tau can play a role in stabilizing the mature axonal cytoskeleton, while the other smaller isoforms might be more associated with axonal growth [22,84]. Big-Tau isoform is, in fact, mainly expressed in neurons projected to the periphery, which, having long or high-caliber axons, need robust and efficient axonal transport. The inclusion of the additional large amino acid segment, specific to the Big-Tau isoform, may determine a significant alteration of the Tau effect on microtubule spacing [62].

Unlike the rest of the Tau protein, which is highly phosphorylated (>80 sites), the insert specific to Big-Tau contains only two phosphorylated sites. This can result in a lower propensity to form toxic aggregates and fibrils than the low-molecular-weight isoforms. Moreover, there is no evidence of homology of the Big-Tau insert with known proteins or functional domains. Taken together, these considerations led some authors to speculate that the big insert of this isoform could be a functionally inert zone that only provides length to the projection domain of Tau and that the evolutionary origin of the insert was from an event of exonization, in which an intronic sequence from a different protein became an exon de novo [62].

### 3.4. Isoforms including Exon 6

Tau isoforms, including exon 6, are preferentially expressed in peripheral tissues, such as skeletal muscle and the spinal cord, but they can also be found at a minor level in some regions of the adult brain and in the fetal brain [40]. Three possible isoforms, 6c, 6p, and 6d, can be generated through the alternative of a different 3′ splice site for E6. Isoform 6c includes the whole sequence of the exon 6 and expands the PRR, increasing the area susceptible to phosphorylation and therefore the possibility of Tau function regulation, albeit a clear consensus on exon 6 function in Tau protein is still missing [26,85,86].

The use of two other alternative 3′ splice sites, 6p and 6d (proximal and distal to the beginning of exon 6, respectively), determines a frameshift that introduces a premature stop codon, thus originating truncated forms of Tau protein represented only by the N-terminal region [40,85,87].

The biological function of these short-Tau isoforms remained completely elusive until recent times [22]. In 2006, Leroy and colleagues identified a brain-specific decrease of the 6c Tau isoform and an increase of the 6d isoform in patients bearing type 1 myotonic dystrophy and described a similar dysregulation of the 6c isoforms in in vitro differentiating neurons treated with retinoic acid [39]. Moreover, Lapointe and colleagues demonstrated that short truncated Tau isoforms (2N6P and 2N6D) were able to block the full-length Tau polymerization in vitro [85]. As fatty acids have been proven to induce Tau aggregate formation [88], the authors used arachidonic acid to induce protein aggregation of various Tau isoforms and demonstrated that not only were these isoforms less prone to form aggregates, but they also inhibited the aggregation of the normal-length Tau protein [85]. The lack of MBTD, which is needed for self-aggregation, can explain why 6p and 6d isoforms do not aggregate and possess anti-aggregative properties [51,85]. The expression pattern of exon 6 is spatially and temporally regulated. The 6p isoforms are the predominant ones, with similar levels in fetal and adult brains, while the 6d level is higher in the fetal brain. Both 6p and 6d can be found in different CNS areas, with the highest levels in the spinal cord and cerebellum [61,85,87]. Particularly, in the cerebellum, the 6d isoforms show levels comparable to those of the full-length isoforms, but in this anatomic region, neurofibrillary tangles are absent, even in patients with neurodegenerative disorders [22,85].

### 3.5. W-Tau

A novel Tau isoform was recently described by Garcia-Escudero et al. [51]. This isoform derives from the TIR-MAPT transcript, which is generated by the retention of intron 12. The TIR-MAPT translation gives rise to a protein with a unique 18-aa sequence, close to the MTBD, and lacking the C-terminal region due to the presence of a premature stop codon in the retained intron. The name “W-Tau” was proposed to indicate the presence of two tryptophan residues, an amino acid that is not present in any other known Tau molecule. This 18-aa sequence specific to W-Tau has also been proposed to trigger a different Tau conformation that elicits the paper-clip state [51].

While other Tau isoforms show a high degree of interspecies homology, W-Tau is human-specific as it contains a sequence translated from an intronic region, which is usually not phylogenetically conserved [22]. The W-Tau expression differs based on brain areas and in the presence of pathological conditions; diminished levels of this new isoform have been found in the brains of Alzheimer’s patients, suggesting a possible role in the pathology. This new Tau isoform exhibits similar post-transcriptional modifications by phosphorylation and affinity for microtubule binding compared to the main Tau isoforms, but more interestingly, it is less prone to aggregate than other Tau isoforms [51]. The reasons behind this decreased aggregation capacity but conservation of microtubule assembly capacity remain unclear and purely speculative. It was suggested that the particular W-Tau domain composition might explain this different behavior. The unique 18-aa extra-peptide includes the sequence GVGWVG, which could be similar in nature to that of some recently described inhibitors of Tau and amyloid β aggregation. In addition, the lack of the C-terminal region implies the loss of a 12-aa sequence after the R4 segment of the MTBD (exon 13), which is found in the core of Tau filaments isolated from the brain of patients affected by some tauopathies, including AD [51].

## 4. Tau and Cancer

A complex relationship between Tau and cancer has emerged in recent years. An increased Tau expression was associated with taxane resistance and higher patient survival in some kinds of cancer but with lower survival in others, while Tau knockdown was associated with tumor proliferation [7,8,89,90,91].

Several findings show that Tau plays a role in multiple functions tightly linked to cancer, such as DNA protection [55,56,92], gene expression [93,94,95], RNA metabolism and protein synthesis [96,97] and miRNA activity [98]. Other works also indicate the involvement of Tau in genome stability [99] and in the regulation of heterochromatin integrity [100]. Cell lines from patients carrying P301L and other *MAPT* mutations present chromatin abnormalities, aneuploidy, and other chromosome aberrations [101,102]. The differential response of mutated and wt cells to microtubule and DNA perturbation has recently been described [103], pointing to Tau as a key player in biological pathways relevant for cancer and confirming that Tau mutations can be considered as cancer risk factors [104].

Survival analysis from data obtained from publicly available datasets showed that high expression of the *MAPT* gene predicted favorable outcomes in ER-positive breast cancer [105]. Breast cancer cell lines and tissue samples confirmed the downregulation of *MAPT* in tamoxifen-resistant patients compared to the sensitive ones. Gene set enrichment analysis strongly linked the *MAPT* gene to immune-related signaling pathways [105]. A possible role for Tau in glioblastoma by controlling 3D cell organization and functions via the PI3K/AKT signaling axis has also been recently suggested by Pagano and colleagues [106]. Depletion of Tau by RNA interference in a 3D model of multicellular spheroids (MCS) showed inhibition of MCS growth and cell evasion and significantly increased median mouse survival in a glioblastoma xenograft model. Furthermore, intracellular signaling array analysis revealed defective activation of the PI3K/AKT pathway in Tau-depleted cells, which was associated with reduced MCS growth and cell evasion. A positive correlation between the amount of phosphorylated Akt-Ser473 and the expression of *MAPT* RNA encoding Tau was also shown by the analysis of the glioblastoma dataset from the Cancer Genome Atlas. The authors, therefore, suggest a role for Tau as an upstream regulator of the PI3K/AKT pathway: Akt activity may be necessary for Tau-dependent cell migration, while upstream PI3K activity may be involved in the control of cell growth and survival, indicating that Tau can contribute to tumor progression, possibly through the regulation of cell growth and/or invasion [106].

Similar results were obtained for breast cancer cells, where decreased Tau protein in lower proliferation, decreased migration, and reduced cell invasion [107]. However, such a function is not shared by all cell types. For instance, in ovarian cancer, the down-regulation of Tau affected the viability of the high Tau-expressing TOV112V cell line while did not affect the low-Tau-expressing OVCAR cell line, suggesting a possible link with the microtubule composition and the level of Tau expression [91,106,108]. Moreover, some authors proposed a direct function of Tau in stabilizing DNA and/or mitotic spindle assembly, conferring a selective advantage for cancer cell growth [99,109].

## 5. Human *MAPT* Mutations Causing Aberrant Tau Splicing in Tauopathies

Tauopathies are a group of heterogeneous disorders characterized by the common feature of accumulation of Tau isoforms in the cytoplasm. They include Alzheimer’s disease (AD), progressive supranuclear palsy (PSP), amyotrophic lateral sclerosis (ALS), Pick disease (PD), corticobasal degeneration, frontotemporal dementia with parkinsonism inked to chromosome 17 (namely FTDP-17), and frontotemporal lobar degeneration with tauopathy (FTLD-Tau) [1,14]. The classification of tauopathies considers both the histopathological features and the predominance in the pathological aggregates of the 3R-Tau or 4R-Tau differing for the exon 10 inclusion/exclusion. Mutations in the human *MAPT* genes have been identified in inherited forms of FTDP-17, establishing that Tau dysfunction can be a direct cause of neurodegeneration. Up to 53 pathogenic *MAPT* mutations have been described until now [45]. Analyzing inherited forms of FTDP-17, mutations were located on exons 9, 10, and 13 and in the exon10/intron 10 junction splice site, highlighting a hot spot of splicing regulation in the intron downstream sequences of exon 10 [45]. Missense, silent, and deletion mutations in E10 have also been shown to affect E10 splicing. Mutations, such as N279K (numeration referred to the longest CNS Tau isoform), empower an exon splicing enhancer (ESE) present in exon 10, whereas a deletion at position 280 (ΔK280) located in the same ESE inhibits E10 inclusion. The amino acid deletion reduces the 4R/3R ration with the opposite effect of the N279K mutation [110]. The mutation N296H/N conversely affects an exonic splicing silencer (ESS), leading to an increase in E10 inclusion [111]. Exon 10 5′ splice-site mutations at positions +3, +13, +14, and +16 are associated with the accumulation of inclusions that contain predominantly Tau 4R isoforms [112,113]. Extending the identification of pathogenic mutations in the frontotemporal dementia (FTD) condition, to date, 112 unique mutations in *MAPT* variants have been identified (https://www.alzforum.org/mutations/mapt, accessed on 2 October 2022). In an interesting recent study, applying human induced pluripotent stem cell (iPSC)-derived cerebral organoids expressing the Tau-V337M mutation in exon 12, the authors highlighted early events that precede the neurodegeneration [114]. Interestingly, they demonstrated that V337M organoids exhibit the expression of glutamatergic signaling pathways and synaptic genes in the early stage and progressively accumulate total and phosphorylated Tau (P-tau S396). The organoid cells showed alteration of autophagy and lysosomal proteins, formation of stress granules, and extensive changes in splicing, followed by the death of glutamatergic neurons. They also give evidence that glutamate-induced cell death in the organoids carrying V337M can be blocked by the lipid PIKFYVE kinase inhibitor apilimod [114]. Investigation of this type of pharmacological inhibitor that prevents glutamatergic cell death may open a new therapeutic approach in FTD.

An alternative animal model for studying Tau mutations is the zebrafish, in which transgenic lines expressing the human 4R Tau isoform display destabilization at the microtubule level with an accumulation of the Tau protein in the cell body [115]. In Zebrafish, the Tau P301L mutation leads to an impaired “touch escape response” due to an altered locomotor system [116], whereas lines expressing hTauA152T show morphological defects such as the curvature of the body [117]. Zebrafish models overexpressing the Tau protein have also contributed to identifying Tau interaction factors and molecules able to counteract the toxicity in neurons induced by Tau mutations [118]. For instance, using the hTauP301L zebrafish line, the Tau-interacting protein FKBP52 (FKBP prolyl isomerase 4) has been suggested to potentially act as a neuroprotective factor [119]. In addition, the same zebrafish hTauP301L line allowed for the identification of heparan sulfate (glucosamine) 3-O-sulphotransferase 2 inhibitors, surfen or oxalyl surfen, as useful molecules to treat neurotoxicity [120,121].

## 6. Alternative Splicing Regulation of Tau Exons

Alternative splicing of pre-mRNA is an essential mechanism of post-transcriptional regulation that enhances the complexity of the information codified by the human genome, contributing to tissue-specific activities and differentiation diversity [122]. It is estimated that almost 95% of human genes may be regulated by alternative splicing [123]. In addition to ubiquitously expressed spliceosomal ribonucleoproteins, tissue-specific alternative splicing factors are required for cell differentiation and tissue specificity [124]. Alternative splicing is regulated by *trans*-acting factors that bind to *cis*-elements located in exonic or intronic sequences [125]. The *trans*-acting factors are mainly represented by two different groups of proteins: the serine–arginine-rich (SR) proteins and RNA-binding proteins, including heterogeneous nuclear ribonucleoproteins (hnRNPs) [126,127]. *Cis*-acting sequences and *trans*-acting factors that participate in Tau exon regulation have been identified as mainly producing Tau minigenes. Table 1 reviews some of the minigene constructs used to model and study Tau alternative splicing patterns. As expected, a great majority of the Tau minigenes currently available model Tau E10 splicing, with E10 being the most characterized Tau alternatively spliced exon [22,26]. Some of these minigenes include only the exonic sequence of E10 [128], while others include intronic sequences flanking E10 itself and containing intron *cis*-acting sequences [129,130]. One of the most commonly used minigenes to study Tau exon 10 AS is the Luc M14 luciferase reporter vector, in which E10 exclusion causes a frameshift in the *P. pyralis* luciferase gene, disrupting the luminescence [131]. This vector, thus, allows a quick determination of E10 exclusion without the need to perform classical RT-qPCR analyses.

Different minigenes have been produced to study E2 and E3 inclusion/exclusion [36,132]. Li and colleagues used different exon 2 minigenes and in situ mutagenesis to demonstrate the presence of a strong splicing enhancer and a weak splicing silencer inside exon 2, and a silencer in the downstream intron [36] regulated during tissue differentiation. Furthermore, a Tau minigene including exon 6 has been described [40].

### Tau Splicing and Ribonucleoproteins

Exon 10 splicing is finely regulated by a tuned balance of *cis*-acting sequences and *trans*-acting factors. Alternative splicing of E10 is tightly regulated by short *cis*-elements present both in the exon and in the downstream intron; those elements act by modulating the use of the weak 5′ and 3′ splice sites of the exon [83]. In fact, at the 5′ end of *MAPT* E10 there are (i) a SC-35-like element whose deletion promotes exon inclusion in cell cultures [136], (ii) a polypurine enhancer (PPE) and an A/C-rich enhancer (ACE) [83], while the 3′ end of E10 harbors both a splicing silencer and an enhancer [26]. As regards the downstream intron (I10), at its 5′ end, an intronic splicing silencer and an intronic splicing modulator compete in order to block or activate the 5′ splice site of the exon [83]. The *cis*-acting sequences expand in the intron and exon sequences, represented primarily by the flanking exons, extension, and sequences in the flanking introns, and the enhancer motif in exon 10. Several SR proteins have been demonstrated to regulate exon 10 expression, including SRSF1, SRSF2, SRSF3, SRSF4, SRSF6, SRSF7, SRSF9, and SRSF11 [83]. By using RNA antisense purification mass spectrometry, Xing and colleagues identified 15 novel factors that bind to Tau pre-mRNA, nine of which play a role in Tau exon 10 splicing [133]. Specifically, they demonstrated that hnRNPC promotes Tau exon exclusion.

It has been demonstrated that somatodendritic Tau may regulate the translational stress response and the biology of ribonucleoproteins (RBPs), leading to the sequestration of RBPs in the cytoplasm [97]. In tauopathy, somatodendritic Tau co-localizes with TIA1 (TIA1 Cytotoxic Granule Associated RNA Binding Protein, also known as T-Cell-Restricted Intracellular Antigen-1), a RBP that regulates alternative pre-RNA splicing and mRNA translation by binding to uridine-rich (U-rich) RNA sequences [137] and increases the formation of stress granules (SGs).

Recently, it has been demonstrated in an AD mouse model that the reduction of TIA1 leads to a delay of degeneration [138] and that the dysregulation of RNA splicing is a relevant pathophysiological mechanism in tauopathies [139].

RBM4, a tissue-specific alternative splicing regulatory factor expressed in the neurons of the hippocampus and frontal cortex, has been demonstrated to participate in the Tau exon 10 inclusion by interacting with an intronic pyrimidine reach element located 100 nucleotides downstream of the 5′ splice site of exon 10 [140]. RBM4 interacts with an RNA helicase, DED/H box polypeptide 5 (DDX5), stimulating Tau exon 10 inclusion. DDX5 interaction with the steam–loop region facilitates U1 snRNP binding to the 5′ splice site of exon 10 [141].

By using chimeric Tau minigene SRp20 (also SRSF3), SRp 30c (also SRSF9), SRp55 (also SRSF6), SRp75 (also SRSF4), 9G8, U2AF, polypyrimidine tract-binding protein (PTBP1), and hnRNP G have been shown to repress Tau exon 10 inclusion. By similar chimeric Tau systems, CELF3 and CELF4 neuronal RBPs have been shown to activate exon 10 splicing [142]. Furthermore, it has been demonstrated that two factors, Tra2β and ASF/SF2, regulate exon 10 splicing by interacting with exon 10 purine-rich enhancer [143]. The hot spot of mutations upstream of the 5′ splice site region in Tau exon 10 is characterized by a stem-loop structure that is required for binding to U1 snRNP (small nuclear ribonucleoprotein) and the splicing factor proline and glutamine rich (SFPQ) [141]. Interestingly, it has been demonstrated that SFPQ accumulates in the cytoplasm in Alzheimer’s and Pick’s disease in brain areas affected by tauopathy and that this cellular localization is mediated by Tau overexpression [144].

At the start of the 2000s, Wei and colleagues showed that overexpression of polypyrimidine tract-binding protein (PTBP1), one of the key regulators of mRNA splicing (reviewed in [145,146,147]), induced Tau 6 exon exclusion in COS cells, while overexpression of U2AF [148] promoted exon 6 retention in the mature mRNA [87]. Strikingly, the authors report that simultaneous overexpression of both proteins yielded the same outcome as overexpression of PTBP1 alone, suggesting that the two RBPs could compete for the same binding sites on the target mRNA [87]. PTBP1 and the neural homologue PTBP2 act in synergy with the corepressors of alternative splicing, Raver 1 and Raver 2, and compete with tissue-specific AS regulatory factors such as RBM20 [145,149,150]. It is not excluded that these factors may participate in the Tau exon splicing regulation. In the following years, many other splicing regulators were tested for their ability to modulate Tau exon 6 AS [39,151]. Based on their effect on exon 6 inclusion/exclusion, RBPs affecting exon 6 splicing can be classified into four major families. The first family comprises RBP able to increase the inclusion of full exon 6 (6+) by decreasing the 6p splicing isoform, such as SRp20 (SRSF3), SRp40 (SRSF5), SRp55 (SRSF6), and SRp75 (SRSF4) [151]. The second family is formed by SC35, hnRNPA1, KSRP, SLM2, YT521B, ASF, SLM1, SRp30c (SRSF9), PTBP1, and htra2b1, which are all able to promote the expression of isoforms 6-, 6p, or 6d [151]. A third family of splicing regulators is composed of RBPs able to decrease both 6+ and 6p isoforms (such as Nova-1) or 6d isoforms (such as 9G8 and SWAP) [151]. The fourth and final family comprehends CELF5 and CELF6, which are both able to promote the 6p Tau isoform expression [39].

The mechanism of the alternative splicing event causing Tau I12 retention is not fully elucidated. However, some evidence suggests that Tau intron 12 retention could be due to the inhibition of GSK3β-mediated modulation of the serine/arginine-rich splicing factor 2 (SRSF2), which was previously linked to intron-modulating splicing events [22]. GSK3β-SRSF2 pathway has also been related to *MAPT*-exon 10 inclusion [152].

RBPs able to influence Tau exons’ splicing are listed on Table 2.

Attempts to regulate Tau exon expression as a therapeutic approach have been undertaken, and some of them have been included in ongoing clinical trials. An antisense oligonucleotide (ASO) approach to induce Tau exon skipping has been demonstrated to reduce Tau aggregation in transgenic mice that express human Tau [153] and has been introduced in clinical trials to target Tau in AD.

**Table 2 ijms-23-15383-t002:** RBPs able to influence Tau exons’ splicing.

RBP	Tau Exon Regulation	Ref.
LUC7L3	Exon 10	[133]
THRAP3	Exon 10	[133]
SRSF1, SRSF2, SRSF3, SRSF4, SRSF6, SRSF7, SRSF9, SRSF11	Exon 10	[24,48,130,154]
EMG1	Exon 10	[133]
PRPF19	Exon 10	[133]
ARL6IP4	Exon 10	[133]
DHX15, 21	Exon 10	[133]
SLM1, SLM2	Exon 6	[151]
hnRNPU, D, A3, HI, C, R, A2B1, A1, G, E2 (PCBP2)	Exon 6, exon 10	[133,151,155]
CLF4, CLF5, CLF6	Exon 6, exon 10	[39]
SC35	Exon6, exon 10	[151,156]
YT521B	Exon 6	[151]
ASF	Exon 6	[151]
Htra2β1	Exon 6, exon 10	[48,129,151]
9G8	Exon 6	[151]
PTBP1	Exon 6, exon 10	[142,151]
PTBP2	Exon 6	[151]
SRSF3, SRSF5, SRSF6, SRSF4, SRSF9,	Exon6	[151]
Swap	Exon 6	[151]
Nova 1	Exon 6	[151]
U2AF	Exon 6	[151]
SFPQ	Exon 10	[141]
RBM4	Exon 10	[140]

## 7. Conclusions

Since the initial characterization of Tau as a microtubule-associated protein that plays a central role in microtubule assembly and cytoskeleton stabilization, it is now accepted that Tau can be considered a multifunctional protein. Evidence demonstrates that Tau is involved not only in the structural integrity of neurons, signal transmission between neurons, and axonal transport but also in the regulation of gene expression, protection of genome architecture, RNA metabolism, and protein synthesis. Abnormal forms of Tau in neuronal tissue are associated with several neurodegenerative diseases known collectively as tauopathies, including Alzheimer’s and FPD. However, the deregulation of Tau isoform expression has also been observed in additional pathological conditions that do not directly affect nervous tissue, such as inclusion body myositis and different types of cancer.

The *MAPT* gene undergoes extensive alternative splicing, producing several isoforms that vary not only for the presence of 1-4R domains but also for the length of the proline-rich region and the N-terminal region, leading to considerable changes in protein dimension.

Regulation of Tau transcript expression has been intensively investigated over the years, and knowledge has increased with recent genomic analyses. Extensive studies have also unraveled the mechanisms of Tau cell-to-cell transfer, such as translocation through the plasma membrane, membranous organelles-based secretion, and ectosomal shedding [157].

The study of the regulatory mechanisms of Tau alternative splicing events has achieved an in-depth understanding of the *cis* sequences and *trans* factors that regulate exon 10. In addition, functional expression studies of the different isoforms of Tau derived from the alternative splicing of exons 2, 3, 4a, 6, and 8 have extended the field of Tau investigation to the tissue-specific role of Tau isoforms.

Clinical investigation of tauopathies, including familial tauopathies, and the development of animal models analyzing the effect of Tau mutations have contributed to a better clinical classification of tauopathies and to the understanding of the pathogenic mechanisms related to altered microtubule functions and aberrant aggregation of Tau protein.

Despite the progress, many aspects still need to be understood: the complex dynamic of Tau fragment aggregation; the role of post-translational modifications, such as acetylation and nitration in addition to phosphorylation; the contribution of the expression of modifier genes and their variants in FPD diseases; and the *trans* factor requirement to maintain the 4R/3R isoform rate [22].

Moreover, in addition to the six major isoforms of the CNS, which have been extensively studied, the functions and distribution of other less-represented Tau isoforms, such as those derived from the AS of exons 4, 6a, and intron 12, deserve attention. Future studies should investigate the functions of isoforms expressed in different tissues and the regulation of their expression, as well as the possible presence of new isoforms derived from other forms of AS.

Deepening insights into splicing factors and drugs that modulate Tau splicing regulation may contribute to elucidating disease pathogenesis and progression and to the development of new treatments for tauopathies.

## Figures and Tables

**Figure 1 ijms-23-15383-f001:**
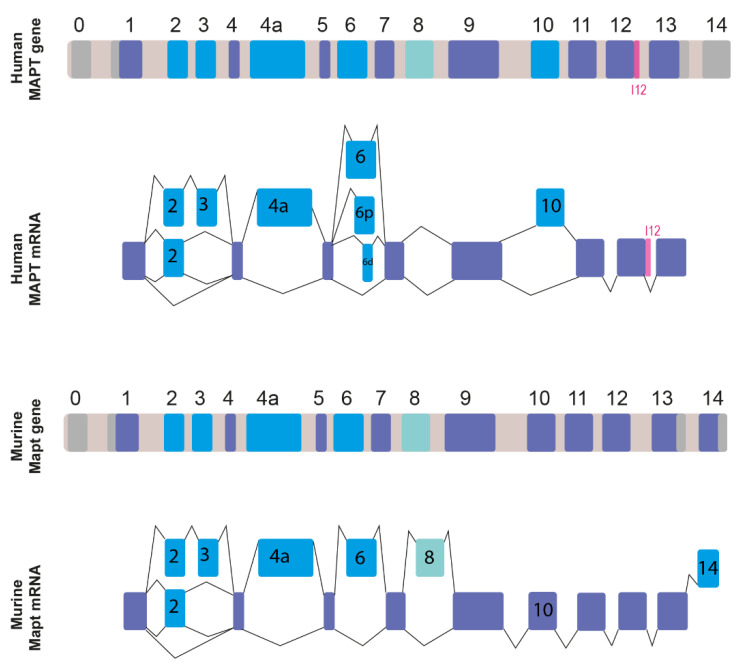
*MAPT* gene organization and alternatively spliced exons in human and mouse transcripts. Untranslated exons (exon 0, the first part of exon 1, the last part of exon 13, and exon 14) are shown in gray. Constitutive exons (the last part of exon 1, exons 4, 5, 7, 9, 11, 12, and the first part of exon 13) are depicted in indigo. Exons that are alternatively spliced (2, 3, 4a, 6, 8, 10) are represented in light blue. The potential retention of intron 12 in human *MAPT* mRNA is highlighted in pink.

**Figure 2 ijms-23-15383-f002:**
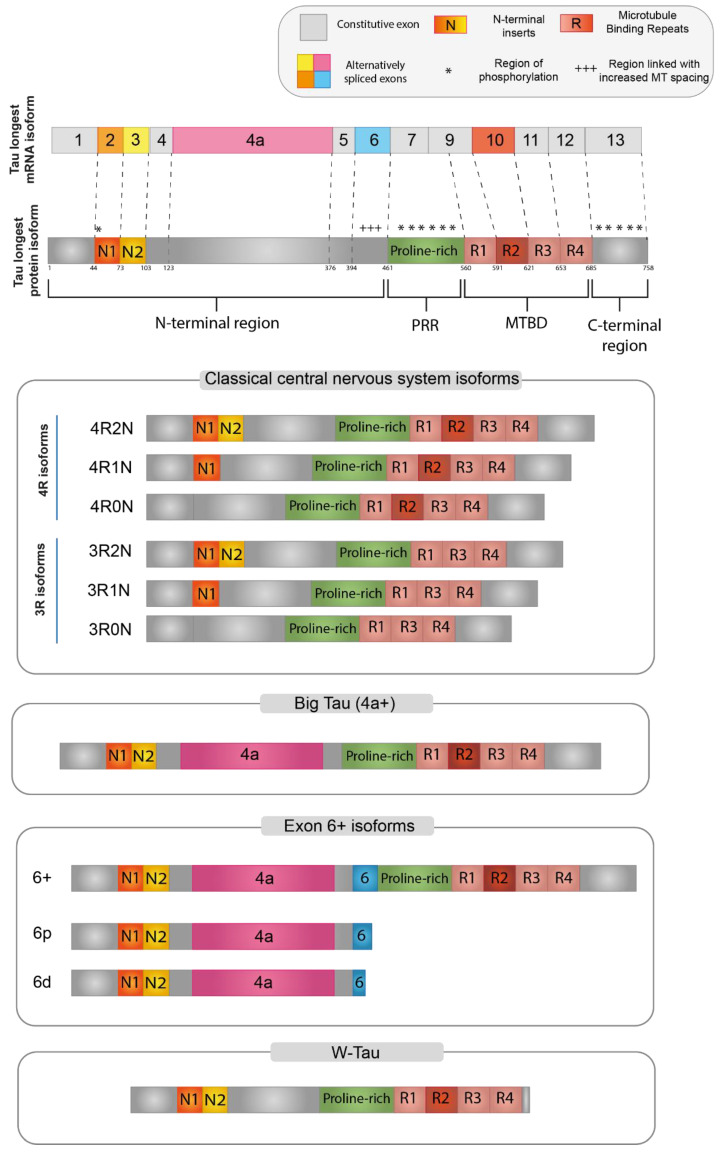
Tau protein isoforms. Schematic representation of the main Tau protein isoforms generated via alternative splicing. At the top, the longest Tau isoform found in humans, containing all the alternatively spliced exons. Red boxes represent the microtubule binding domain (MTBD), while the N-terminal projection domain is represented by exon 2 (orange) and exon 3 (yellow). For Big Tau (4+), Exon 6+ isoforms and W-Tau only the 2N4R isoforms are depicted, but all combinations are possible.

**Table 1 ijms-23-15383-t001:** Mini-reporter genes for Tau exons splicing evaluation.

Minigene Name	Exon(s)	Properties	Application	Ref.
hTau	1, 2, 4	Exon 2 flanked by different length of its native introns, surrounded by E1 and E4	Identification of *cis* and *trans* acting elements altering splicing of E2	[132]
SV1/2L/4	2	Exon 2 flanked by its native exons	In situ mutagenesis to identify *cis* acting elements in the exon and downstream intron	[36]
SP/2L	2	Exon 2 flanked by heterologous exons	In situ mutagenesis to identify *cis* acting elements in the exon and downstream intron	[36]
SVΔ2/3	3	E3, 1.1 kbps of I2 and 1.4 kpbs of I3	E3 default splicing pattern analysis, identification of *cis*-acting elements	[37]
pSVIRB SV6	6	E6, 600 bps I5, 200 bps I6	Exon-trapping assay	[40]
TauEx9-11d5	9-10-11	E10 flanked by 269 and 246 bp of its native intron and exons 9–11.	Identification of *cis*-acting elements altering splicing of E10	[129]
SI9/SI10 LI9/SI10 SI9/LI10 LI9/LI10	9-10-11	E10 can be flanked by a long (L) or short (S) portion of the upstream and downstream intron	Identification of a required minimal distance for E10 splicing	[130]
Tau minigene	9-10-11		Identification of HnRNPC as a novel *trans*-acting factor on E10	[133]
RHCGlo Tau wt RHCGlo Tau Δ280K RHCGlo Tau N279K	10	Contains WT E10 or E10 with Δ280K, and N279K mutations	Identification of *cis* and *trans*-acting elements affecting E10 splicing	[128]
Luc M14	10	Luciferase reporter, modified from TauEx9-11 d5 minigene	Identification of compound that reduced exon 10 inclusion; identify *cis*/*trans* elements	[130,131,134,135]

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
