# Peer review of "Tau Isoforms: Gaining Insight into MAPT Alternative Splicing"

_ijms, 2022, doi:10.3390/ijms232315383_

Round 1
Reviewer 1 Report
This is an ample review about Tau expression, models for the role of cis - elements and ribonucleoproteins for regulating alternative splicing of Tau exons.
Because the review adds a multitude of descriptive data on the mentioned topic, a longer and more functionally oriented conclusion would help the reader to bring together the wealth of information in a concise manner.
Minor errors:
- line 9. After the Abstract, the authors should delete the directive for writing.
line 266. Please delete: Error Bookmark not defined
Author Response
#1 Reviewer
This is an ample review about Tau expression, models for the role of cis - elements and ribonucleoproteins for regulating alternative splicing of Tau exons.
Because the review adds a multitude of descriptive data on the mentioned topic, a longer and more functionally oriented conclusion would help the reader to bring together the wealth of information in a concise manner.
We thank the Reviewer for the valuable time and comments that have contributed to improving the manuscript.
Following the Reviewer’s suggestions, we have extended the conclusion section introducing a concise summary of the content of the reviewer focusing on Tau function.
Minor errors:
- line 9. After the Abstract, the authors should delete the directive for writing.
We deleted the directive for writing
line 266. Please delete: Error Bookmark not defined
We deleted “Error Bookmark not defined”
#2 Reviewer
The manuscript proposed by Dr. Corsi and colleagues describes the molecular mechanisms involved in the complex alternative splicing of the MAPT gene. The resulting review is competently written and up to date. The subject was dealt with concisely, clearly and to the point, despite its intrinsic intricacy. The only flaw concerns paragraph 5.1 on the transgenic zebrafish model, which is detached from the rest of the paper in terms of both writing style and contents. It does not add much and should be rewritten or even deleted.
Following the reviewer's suggestions, we removed paragraph 5.1 and selected part of the content of the paragraph, rewritten in a style uniform with the rest of the document, which was introduced in section 5.
As a minor point, the ribonucleoprotein part is only one of several aspects covered and, therefore, the title of the article should not focus on this.
As suggested by the Reviewer we have modified the title as follows “Tau isoforms: gaining insight into MAPT alternative splicing”
Reviewer 2 Report
The manuscript proposed by Dr. Corsi and colleagues describes the molecular mechanisms involved in the complex alternative splicing of the MAPT gene. The resulting review is competently written and up to date. The subject was dealt with concisely, clearly and to the point, despite its intrinsic intricacy. The only flaw concerns paragraph 5.1 on the transgenic zebrafish model, which is detached from the rest of the paper in terms of both writing style and contents. It does not add much and should be rewritten or even deleted.
As a minor point, the ribonucleoprotein part is only one of several aspects covered and, therefore, the title of the article should not focus on this.
Author Response
#2 Reviewer
The manuscript proposed by Dr. Corsi and colleagues describes the molecular mechanisms involved in the complex alternative splicing of the MAPT gene. The resulting review is competently written and up to date. The subject was dealt with concisely, clearly and to the point, despite its intrinsic intricacy. The only flaw concerns paragraph 5.1 on the transgenic zebrafish model, which is detached from the rest of the paper in terms of both writing style and contents. It does not add much and should be rewritten or even deleted.
We thank the Reviewer for the valuable time and comments that have contributed to improving the manuscript.
Following the reviewer's suggestions, we removed paragraph 5.1 and selected part of the content of the paragraph, rewritten in a style uniform with the rest of the document, which was introduced in section 5.
As a minor point, the ribonucleoprotein part is only one of several aspects covered and, therefore, the title of the article should not focus on this.
As suggested by the Reviewer we have modified the title as follows “Tau isoforms: gaining insight into MAPT alternative splicing”
Round 2
Reviewer 1 Report
Now, the text is better understandable, especially by the addition in Conclusion.
Also the comparison with the Zebrafish model (in Discussion) brings more clarity.